

# Identification and serological responses to a novel *Plasmodium vivax* merozoite surface protein 1 (*Pv*MSP-1) derived synthetic peptide: a putative biomarker for malaria exposure

Aline Marzano-Miranda[1], Gustavo Pereira Cardoso-Oliveira[1], Ingrid Carla de Oliveira[1], Luiza Carvalho Mourão[1], Letícia Reis Cussat[1], Vanessa Gomes Fraga[1], Carlos Delfin Chávez Olórtegui[2], Cor Jesus Fernandes Fontes[3], Daniella Castanheira Bartholomeu[1] and Erika M. Braga[1]

[1] Department of Parasitology, Universidade Federal de Minas Gerais, Belo Horizonte, MG, Brazil
[2] Department of Biochemistry and Immunology, Universidade Federal de Minas Gerais, Belo Horizonte, MG, Brazil
[3] School of Medical Sciences, Universidade Federal de Mato Grosso, Cuiabá, Mato Grosso, Brazil

Corresponding author
Erika M. Braga, embraga@icb.ufmg.br

## ABSTRACT

**Background.** The integration of diagnostic methods holds promise for advancing the surveillance of malaria transmission in both endemic and non-endemic regions. Serological assays emerge as valuable tools to identify and delimit malaria transmission, serving as a complementary method to rapid diagnostic tests (RDT) and thick smear microscopy. Here, we evaluate the potential of antibodies directed against peptides encompassing the entire amino acid sequence of the *Pv*MSP-1 Sal-I strain as viable serological biomarkers for *P. vivax* exposure.

**Methods.** We screened peptides encompassing the complete amino acid sequence of the *Plasmodium vivax* Merozoite Surface Protein 1 (*Pv*MSP-1) Sal-I strain as potential biomarkers for *P. vivax* exposure. Here, immunodominant peptides specifically recognized by antibodies from individuals infected with *P. vivax* were identified using the SPOT-synthesis technique followed by immunoblotting. Two 15-mer peptides were selected based on their higher and specific reactivity in immunoblotting assays. Subsequently, peptides p70 and p314 were synthesized in soluble form using SPPS (Solid Phase Peptide Synthesis) and tested by ELISA (IgG, and subclasses).

**Results.** This study unveils the presence of IgG antibodies against the peptide p314 in most *P. vivax*-infected individuals from the Brazilian Amazon region. *In silico* B-cell epitope prediction further supports the utilization of p314 as a potential biomarker for evaluating malaria transmission, strengthened by its amino acid sequence being part of a conserved block of *Pv*MSP-1. Indeed, compared to patients infected with *P. falciparum* and uninfected individuals never exposed to malaria, *P. vivax*-infected patients have a notably higher recognition of p314 by IgG1 and IgG3.

## INTRODUCTION

Malaria transmission in Latin America and the Caribbean has undergone a marked reduction over the preceding two decades. However, a considerable population of 138 million individuals remains at risk of infection (*WHO, 2023*). *Plasmodium vivax* constitutes a predominant species responsible for 72% of the 552.000 annual malaria cases reported in the Americas imposing a burden on healthcare systems and economies (*WHO, 2023*). Control and elimination of *P. vivax* represent a significant challenge due to its biological characteristics including the occurrence of relapses and the early circulation of gametocytes (*Adekunle et al., 2015*; *Markus, 2015*; *Recht et al., 2017*; *Commons et al., 2020*; *Merrick, 2021*). The current diagnostic tests based on parasite microscopic detection exhibit limited sensitivity, primarily attributed to the reduced peripheral density of *P. vivax* blood stages correlated to the parasite's preference for young red blood cells, which are more abundant in the spleen and bone marrow (*Harris et al., 2010*; *Mathison & Pritt, 2017*; *Ashley, Pyae Phyo & Woodrow, 2018*; *Kho et al., 2021*). Furthermore, the management of clinical *P. vivax* infections inherently overlooks asymptomatic carriers, significantly contributing to the dynamics of malaria transmission (*Corran et al., 2007*; *Bousema & Drakeley, 2011*; *Da Silva-Nunes et al., 2012*; *Vallejo et al., 2016*; *Longley et al., 2017*; *Tadesse et al., 2017*).

The implementation of accurate diagnostic tools followed by a specific treatment is mandatory to achieve the goals and targets of the Global Technical Strategy for Malaria established by the World Health Organization (*WHO, 2023*). Enhancing the surveillance of malaria transmission in both endemic and non-endemic areas may be achieved through the integration of more sensitive diagnostic measures. Serological assays represent a promising approach because they can determine malaria transmission undetected by routine field diagnostics like RDT and microscopy (*Cutts et al., 2014*; *Longley et al., 2022*; *O'Flaherty, Roe & Fowkes, 2022*).

Malarial infection elicits high antibody titers, mostly against the parasite's blood stage, and those molecules persist even after treatment (*Braga, Fontes & Krettli, 1998*; *Morais et al., 2005*; *Costa et al., 2020*). Individuals living in endemic areas for *P. vivax* swiftly develop considerable titers of antibodies towards specific parasite antigens (*Bueno et al., 2011*; *Fantin et al., 2021*; *França et al., 2021*; *Punnath et al., 2021*; *Tayipto et al., 2022*). The identification of antibodies that target *P. vivax* antigens holds utility in delineating vivax malaria transmission regions, pinpointing asymptomatic reservoirs, predicting outbreaks, and facilitating the effective deployment of control measures, including prompt treatment (*Greenwood et al., 2008*; *Kim et al., 2014*; *Longley et al., 2017*; *O'Flaherty, Roe & Fowkes, 2022*).

Among several blood-stage antigens, the *P. vivax* Merozoite Surface Protein 1 (*Pv*MSP-1) emerges as a potential diagnostic target due to its role in erythrocyte invasion by merozoites (*Holder & Freeman, 1982*; *Del Portillo et al., 1991*). This approximately 195 kDa protein, synthesized by schizonts and expressed on merozoite surface, is well recognized by *P. vivax*-infected patients, eliciting a robust immune response (*Braga, Fontes & Krettli, 1998*; *Soares et al., 1999*; *Morais et al., 2005*; *Scopel et al., 2005*; *Pandey et al., 2010*; *Mourão et al., 2012*; *Riccio et al., 2013*; *Wang et al., 2016*; *Rocha et al., 2017*; *McCaffery et al., 2019*).

Therefore, the identification of the immunogenic targets in *Pv*MSP-1 that elicit a significant antibody response could help the identification of new biomarkers for malaria exposure.

Here, we evaluate the potential of antibodies directed against peptides encompassing the entire amino acid sequence of the *Pv*MSP-1 Sal-I strain (*Putaporntip et al., 2002*) as viable serological biomarkers for *P. vivax* exposure. We employed SPOT-synthesis and epitope mapping techniques to identify the immunodominant epitopes exclusively recognized by antibodies from *P. vivax*-infected patients. A specific peptide (p314) displays high reactivity and selectivity for antibodies from *P. vivax*-infected individuals who reported a brief exposure to malaria in the Amazon Basin endemic areas. Overall, this study introduces a *P. vivax*-specific peptide, representing a promising candidate for inclusion in future serological biomarkers to measure exposure to vivax malaria.

## MATERIAL AND METHODS

### Study population

This study enrolled serum samples from 156 adult *P. vivax*-infected patients aged 18 to 68 years (median age: 38 years). The majority of the samples (75%) were obtained from male subjects between February 2006 and January 2008. Patients were selected based on classic malaria symptoms (fever, chills, headache, myalgia, nausea, and vomiting), seeking medical attention at Hospital Universitário Júlio Muller in Cuiabá (HUJM-CB), Mato Grosso State, Brazil. Notably, active malaria transmission does not occur in Cuiabá. Patients reported short exposure to other areas in the Brazilian Amazon where malaria is endemic, suggesting potential infection acquisition during these visits. Indeed, the cumulative exposure in the endemic regions was limited to less than one year, exhibiting a median incidence of three previous malaria cases. Our analysis also included 15 patients with *P. falciparum* infection, selected based on identical criteria at HUJM-CB during the same period. All patients were diagnosed by thick blood smear microscopic observation and the infection was confirmed by a species-specific nested PCR amplification of the *Plasmodium* spp. 18S SSU rRNA gene (*Scopel et al., 2004*). As controls, 36 malaria-naïve individuals residing in a non-endemic area (Belo Horizonte, state of Minas Gerais, Brazil) with no prior exposure to malaria transmission were also included. Blood was collected by vein puncture into EDTA tubes, plasma samples were separated by centrifugation (520× g for 15 min) and stored at $-20\,^\circ C$ until serological experiments were conducted. The sera tested comprise a Biorepository at the Malaria Laboratory/ Universidade Federal de Minas Gerais (UFMG), adhering to Ethics Committee of the National Information System on Research Ethics Involving Human Beings regulations (SISNEP–CAAE: 01496013.8.0000.5149). All individuals included in this study were anonymized and provided written informed consent for collecting samples and subsequent analysis. A detailed description of the characteristics of *P. vivax*-infected patients has been published elsewhere (*Morais et al., 2005*; *Mourão et al., 2012*; *Mourão et al., 2016*).

### Peptide SPOT-synthesis and immunodetection of reactive peptides

A total of 580 overlapping 15-mer peptides, frame-shifted by three residues, covering the entire amino acid sequence of *Pv*MSP-1 Sal I strain (Gene PVX 099980) (*Putaporntip et al.,*

*2002*) were simultaneously synthesized using an automatic synthesizer ResPep SL (Intavis) according to the SPOT-synthesis technique (*Frank, 2002*). Briefly, the peptides were built on specific spots on a derivatized cellulose membrane (Intavis), using Fmoc chemistry. The Fmoc group was removed with 4-methylpiperidine 25% v/v in dimethylformamide (DMF; Merck, Rahway, NJ, USA). The amino acids were activated using diisopropylcarbodiimide (DIC; Sigma-Aldrich, St. Louis, MO, USA) and Oxyma Pure (Sigma-Aldrich) at 1.1 M (1:1) and deposited onto the membrane for coupling. After, non-reactive amine residues were acetylated with acetic anhydride (3% v/v in DMF). These steps were repeated until all the amino acids were added. At the end of the synthesis, the membrane was submerged for one hour in a cleavage solution composed of 95% (v/v) trifluoracetic acid (TFA; Sigma Aldrich) associated with 2.5% (v/v) water and 2,5% (v/v) triisopropylsilane (TIPS; Merck) to remove protective groups from the amino acids side chains. After, the membrane was washed 4 times with dichloromethane (DCM; Sigma-Aldrich), 4 times with DMF, and 2 times with ethanol (Sigma-Aldrich). The membrane was dried and the spots were checked with UV light.

Immunoblotting assays were performed using pools of plasma from *P. vivax*-infected individuals ($n = 10$), *P. falciparum*-infected patients ($n = 10$), and healthy donors ($n = 10$). Briefly, the membrane was blocked overnight with PBS containing 3% (w/v) bovine serum albumin (BSA; Sigma-Aldrich) and 5% (w/v) sucrose. The blocked solution was removed, and the membrane was washed three times with PBS with 0.1% Tween (PBST) with agitation for 10 min each. Antibodies from each pool diluted 1:1000 in PBST were added to the membrane, one at a time, for 2 h at room temperature, under vigorous stirring. After each incubation, the membrane was rewashed three times with PBST 0.1% for 10 min. Antibody binding was detected using HRP-conjugated anti-human IgG (Sigma-Aldrich) diluted 1:10000 on PBST, and was incubated for 1 h at room temperature (23 °C) following three washes with PBST for 10 min. Luminata (Millipore/Merck, Rahway, NJ, USA) was added to the membrane as a substrate. The membrane was automatically exposed, scanned, and digitalized by an ImageQuant LAS 500 chemiluminescence detector (GE Healthcare, Chicago, IL, USA).

## Membrane analysis and spot quantification

The spot intensity of each peptide was quantitatively measured by densitometry scanning analysis using the software Image J–Protein Array Analyzer plug-in (http://image.bio.methods.free.fr/ImageJ/?Protein-Array-Analyzer-for-ImageJ.html) as described elsewhere (*Reis-Cunha et al., 2022*). Briefly, densitometry values were normalized and the mean densitometric value of the four less reactive peptides from each experiment was subtracted from the densitometric value of each spot of the same membrane. The cut-off value was defined based on the mean plus two times the standard deviation of the reactivity of all spots in the membrane incubated with non-infected pool sera.

## *In silico* peptide validation

B cell linear epitopes were predicted *in silico* using the program BepiPred 2.0 (https://services.healthtech.dtu.dk/services/BepiPred-2.0/) as described before (*Bueno et al., 2011*).

Briefly, based on a single FASTA sequence input each amino acid receives a prediction score based on a random forest algorithm trained on epitopes annotated from antibody-antigen protein structures. The cut-off value of 0.5 was used to validate a linear B cell epitope. Additionally, the amino acid sequence was analyzed using the Protein BLAST® (https://blast.ncbi.nlm.nih.gov/Blast.cgi) to determine the specificity of those sequences for *P. vivax*.

## Synthesis of soluble peptides

The 15-mer peptides selected by immunoblotting assays were synthesized on a 10 μmol scale by SPPS technique using the automatic synthesizer ResPep SL (Intavis). Briefly, the Fmoc protective group was removed using 25% 4-methylpiperidine (v/v in DMF), and amino acids were activated with a 1:1 solution of Oxyma Pure (Merck) and DIC (Sigma-Aldrich). The amino acids were incorporated into an H-Rink Amide ChemMatrix resin (Sigma-Aldrich) and non-reagent amino groups were acetylated with acetic anhydride (3% v/v in DMF). These steps were repeated until the synthesis of each peptide was fully completed. Peptides were released from the resin using a cleavage cocktail as previously described (*Siqueira, 2023*). Briefly, the peptides were precipitated by adding cold methyl tert-butyl ether and then lyophilized. We performed a MALDI-TOF spectrometry to check each synthesized peptide's mass-to-charge ratio (m/z) H+ using a saturated matrix solution containing 10 mg/mL α-cyano-4-hydroxycinnamic. The peptide: matrix mixture (1:2) was applied on an MTP AnchorChip™ 600/384 plate (Bruker Daltonics) and maintained to dry at room temperature.

## Serological assays

IgG and IgG subclass antibody responses against the selected peptides were detected by enzyme-linked immunosorbent assay (ELISA). Briefly, each well of a 96-well flat-bottomed polystyrene microplate (Corning Incorporation, Corning, NY, USA) was coated with 0.5 μg/μL of peptide in bicarbonate–carbonate buffer (pH 9.6; 0.1 M) and incubated overnight at 4 °C. Plates were blocked with BSA 3% (w/v) in PBS pH 7.4, for 1 h at 37 °C. Then, each plasma diluted 1:100 in PBST 0.05% containing 3% BSA was tested in duplicate and incubated for 2 h at 37 °C following four washes with PBST 0.05%. HRP-conjugated anti-human (Sigma-Aldrich) IgG (1:2000) or anti-human IgG1, IgG2, IgG3, and IgG4-biotin antibodies were diluted (1:500) in 0.05% PBST with 3% BSA and incubated for 90 min at 37 °C. For IgG subclass detection, HRP-conjugated streptavidin (RD systems, Minneapolis, MN, USA) (1:1000) diluted in 0.05% PBST was added for 60 min at 37 °C. Antibody reaction was revealed using 0.5 mg/mL o-phenylenediamine dihydrochloride (OPD) substrate (Sigma-Aldrich) diluted in 0.05 M phosphate-citrate buffer pH 5.0 and the reaction was stopped with 4M $H_2SO_4$. The optical density (OD) value at 492 nm was determined in a Multiskan GO Reader (Thermo Fisher Scientific, Waltham, MA, USA) microplate spectrophotometer. Levels of IgG and IgG subclasses were expressed as reactivity index (RI). To establish a cut-off, the mean OD was calculated by adding three standard deviations of samples from 10 naïve volunteers who had no previous exposure to malaria. The RI was calculated as a ratio between the mean OD of each patient's duplicate sample by the cut-off value. Any reactivity above 1 was considered responsive.

## Statistical analysis

Data were analyzed using GraphPad Prism 8.0 software (GraphPad Software, La Jolla, CA, USA). To assess if the data fit in a Gaussian distribution, we used the Shapiro–Wilk normality test. Comparisons between groups were analyzed by the Kruskal–Wallis test followed by Dunn multiple comparison tests. Receiver operating characteristic (ROC) analysis was performed to obtain sensibility, specificity, and accuracy values for IgG, IgG1, and IgG3 serological responses against p314. We used the Principal Component Analysis (PCA), an effective technique used to reduce the dimensions of a dataset by replacing the original variables with a smaller set of derived variables (principal components, PCs), to analyze the correlation between IgG subclass levels, malaria exposure, and parasitemia. To accomplish this, we assessed the proximity of position and value of loadings contribution to each variable (*Zhang & Castelló, 2017*; *Beattie & Esmonde-White, 2021*). This analysis was performed in the R software (4.3.0 version) using factoextra (https://cran.r-project.org/web/packages/factoextra/index.html), ggplot2 (https://cran.r-project.org/web/packages/ggplot2/index.html) and MASS (https://cran.r-project.org/web/packages/MASS/index.html) packages. A *p*-value <0.05 was considered statistically significant.

## RESULTS

### *Pv*MSP-1 peptides are exclusively recognized by antibodies from *P. vivax*-infected patients

We employed epitope mapping to pinpoint the immunodominant epitopes within *Pv*MSP-1 exclusively recognized by antibodies from *P. vivax*-infected patients. The peptide recognition patterns by antibodies from each examined group are illustrated in Fig. 1A, revealing that antibodies from *P. vivax*-infected individuals recognize a greater number of peptides in comparison to other groups. We further examined whether the antibody response exhibited variations against the peptides constituting the conserved and polymorphic blocks of *Pv*MSP-1 following the categorization by *Putaporntip et al. (2002)*. Antibodies from *P. vivax*-infected patients notably exhibited a higher reactivity to both conserved and polymorphic blocks than antibodies from individuals of other groups (Fig. S1). A Venn diagram (Fig. 1B) illustrates the peptide recognition by antibodies from each studied group, depicting 255 peptides (53.5%) recognized by antibodies from subjects of the three groups. Of the peptides recognized by a single group, *P. vivax* patients recognized 142 peptides (29.8%); non-infected individuals, 11 peptides (2.3%); and *P. falciparum*-infected patients, 8 peptides (1.7%). Figure 1C depicts the twelve most reactive peptides exclusively recognized by antibodies from *P. vivax*-infected patients. Among these, we selected two, p70 and p314, for further investigation due to their highest relative intensity values. Furthermore, *in silico* analysis revealed six/seven amino acids as B-cell epitopes, LKISDK, and DELDLLF for p70 and p314, respectively (Fig. 2). Concerning the homology between p314 and p70 and other FASTA sequences available on the NCBI website for *Plasmodium* spp., our analysis revealed that the p314 sequence exhibited alignment solely with the *Pv*MSP-1 (Table S1), thereby confirming the specificity of this peptide. Conversely,

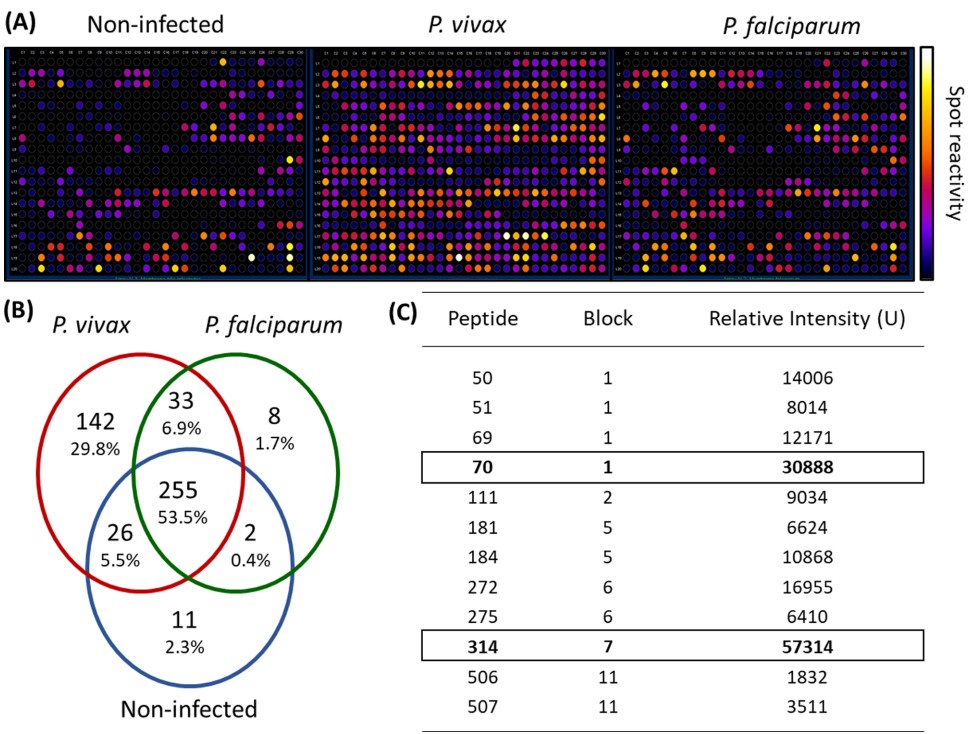

**Figure 1 Antibody interactions with a cellulose-bound peptide library representing the PvMSP-1.** An array of 580 overlapping 15-mer peptides was synthesized on a cellulose membrane. (A) Images of the membrane depict the reactivity at each spot after probing with three distinct pools of sera ($n = 10$): non-infected individuals never exposed to malaria, *P. vivax*-infected patients, and *P. falciparum*-infected patients. (B) A Venn diagram illustrates the number and percentage of reactive peptides for each pool of sera. (C) The box highlights the 12 most reactive peptides exclusively recognized by *P. vivax*-infected individuals.

the p70 sequence exhibited alignment not only with the *Pv*MSP-1 but also with homologous proteins of *P. coatneyi* and *P. cynomolgi*, non-human primate species occurring in Southeast Asia (*Eyles, Coatney & Getz, 1960*; *Ta et al., 2014*; *Zhang et al., 2016*) (Table S1).

## Peptide p314 is highly recognized by IgG from *P. vivax*-infected patients

IgG responses to p70 and p314 were assessed by ELISA and results were expressed as reactivity index. Antibody response against p70 exhibited median values below the cut-off value in all studied groups (Fig. 3A). On the other hand, *P. vivax*-infected individuals exhibited higher levels of IgG against p314 compared to healthy individuals ($p = 0.001$) and those infected with *P. falciparum* ($p = 0.0001$) (Fig. 3B). Based on these results, we further investigate IgG subclass responses only against p314.

## Cytophilic IgG subclasses against p314 are higher in vivax malaria and correlate with parasitemia

Our findings indicate that among the IgG subclasses targeting p314 in plasma from *P. vivax*-infected individuals, IgG3 predominates, followed by IgG1 (Fig. 4). IgG1 and

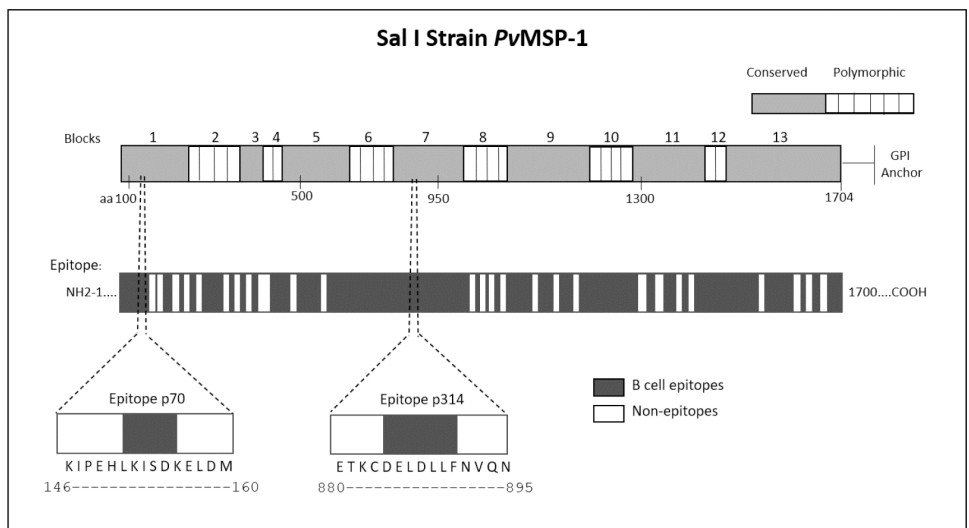

**Figure 2 Schematic representation of the *Pv*MSP-1 structure in amino acids, organized into blocks regions, and predicted epitope regions for B cells.** The first bar illustrates the complete amino acid sequence of the protein, with the numbers above indicating polymorphic and conserved blocks of *Pv*MSP-1. The second bar denotes *in silico* predicted B-cell epitope regions. Additionally, the diagram illustrates the location of p70 and p314 on *Pv*MSP-1, along with their respective sequences and predicted epitopes.

IgG3 levels were significantly three times higher in individuals infected with *P. vivax* compared to non-infected and *P. falciparum*-infected patients. Statistically significant differences were observed between subjects with vivax malaria and the other studied groups (IgG1 $p < 0.0001$, IgG3 $p < 0.001$). However, IgG2 and IgG4 exhibited median values for all groups below the positivity threshold (Fig. S2).

Aiming to pinpoint the predominant anti-p314 IgG subclass among 120 positive *P. vivax* patients, a Venn diagram was created. The distribution of anti-p314 IgG subclasses indicated that IgG1 was positive in 69 patients (57.5%), IgG2 in 52 (43.3%), IgG3 in 102 (85%), and IgG4 in 20 (16.7%) (Fig. 5A). Among those 120 patients, four (3.3%) individuals were positive only for IgG1, 11 (9.2%) only for IgG2, 24 (20%) only for IgG3, and one (0.8%) only for IgG4. Altogether, 80 patients (66.7%) displayed a combination of at least two IgG subclasses, with the most prevalent being a mixed response of IgG1 and IgG3 in 64 ones (53.3%). Next, we correlated the levels of IgG subclass responses with parasite burden and malaria exposure (number of previous malaria episodes). Our findings indicate that IgG3, closely trailed by IgG1, emerges as the predominant factor strongly linked to the parasitemia. Notably, these factors display contrasting trends compared to individuals without malaria infection (Fig. 5B).

## The ROC analysis reveals a similar pattern between anti-p314 IgG and IgG3 responses

To assess the efficacy of anti-p314 antibodies, we analyzed Receiver Operating Characteristic (ROC) curves, evaluating the sensitivity, specificity, and accuracy of IgG, IgG1, and IgG3 immunoglobulins in our ELISA protocol (Fig. 6A). IgG anti-p314 exhibited a specificity

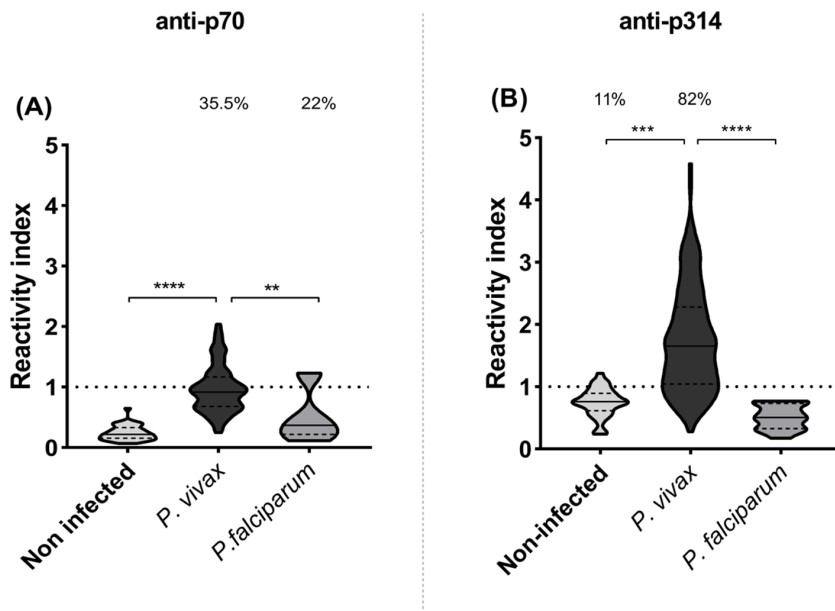

**Figure 3   IgG levels against p70 and p314.** Antibodies levels against soluble peptides were assessed by ELISA using plasma from non-infected individuals ($n = 36$), *P. vivax* ($n = 156$), and *P. falciparum* patients ($n = 15$). (A) IgG levels anti-p70; and (B) IgG levels anti-p314 in plasma from individuals of each studied group. Antibodies levels were expressed as RI. The violin plot indicates interquartile intervals and median. Percentage values indicate the overall frequency of positive responders within each group. Significance was assessed by the Kruskal-Wallis test followed by Dunn's *post-hoc* test. ** $p < 0.01$; *** $p < 0.001$; **** $p < 0.0001$.

of 76.47% and a sensitivity of 77.78% in distinguishing *P. vivax*-infected individuals from those never exposed to malaria parasites (Fig. 6B). For IgG1, we observed a specificity of 49.11% and a sensitivity of 48.39%. In the case of IgG3 anti-p314, the analysis indicates a sensitivity of 77.30% and a specificity of 78.57%. Considering these results, our findings highlight a well-defined antibody response against p314 characterized by IgG and IgG3 (Fig. 6B).

## DISCUSSION

The use of epidemiological surveillance tools associated with the implementation of control measures, such as rapid and accurate diagnosis and adequate treatment, is essential to decrease *P. vivax* transmission. In this study, we assessed the potential of antibodies targeting peptides spanning the entire amino acid sequence of the *Pv*MSP-1 Sal-I strain as viable biomarkers for *P. vivax*'s exposure. For the first time employing the SPOT-synthesis technique along with epitope mapping of *Pv*MSP-1, we identified immunodominant epitopes recognized exclusively by antibodies from *P. vivax*-infected patients. Notably, a specific peptide (p314) exhibits elevated reactivity and selectivity on ELISA assays for antibodies from individuals infected with *P. vivax*.

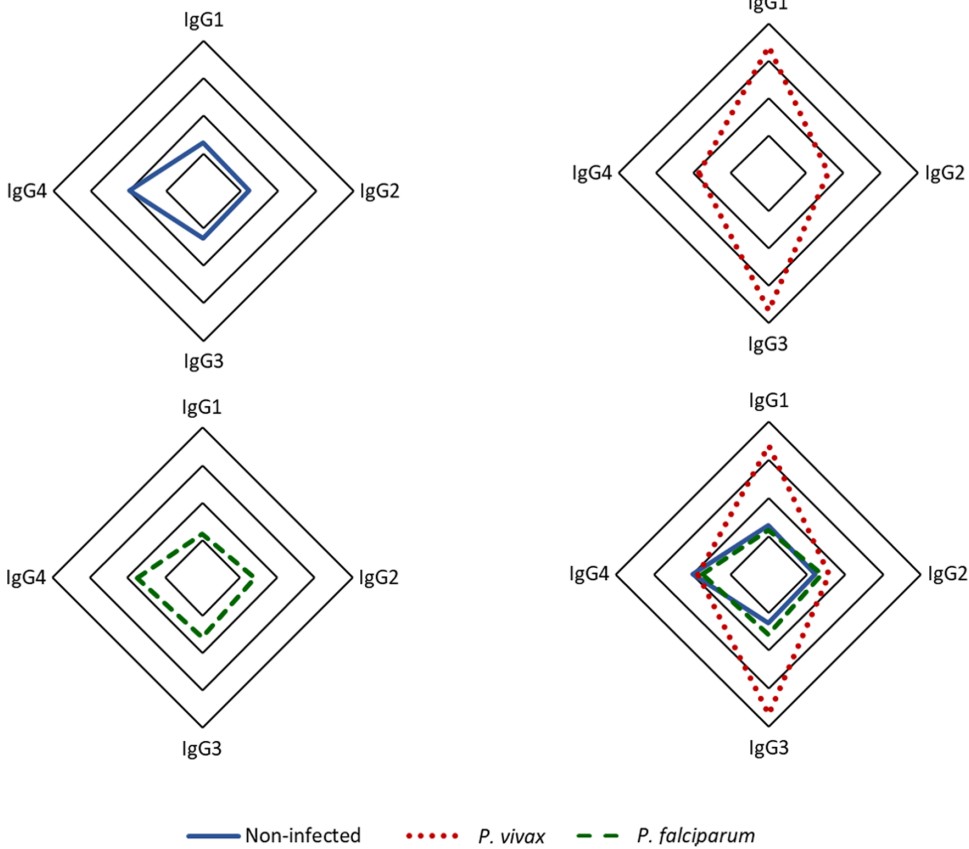

**Figure 4  Distribution of IgG subclasses recognizing the antigen p314.** The radar chart shows the frequencies of each IgG subclass among individuals from the three groups. Panels (A), (B), and (C) represent responses among healthy donors ($n = 35$), *P. vivax* patients ($n = 152$), and *P. falciparum* patients ($n = 35$), respectively, while panel (D) displays the merged groups. *Cut-off* values for IgG subclasses (1-4) are 0.17, 0.31, 0.45, and 0.05, respectively. The radar charts are segmented into four lines, each one representing 25% of the value.

Our findings from the SPOT-synthesis assay indicate that *Pv*MSP-1 possesses 142 specific-linear antigenic determinants recognized by antibodies from individuals infected with *P. vivax*. Patients diagnosed with vivax malaria exhibit heightened IgG responses, targeting distinct regions of the *Pv*MSP-1. Such humoral responses target both conserved and polymorphic sections, as elucidated by a prior characterization of the comprehensive molecular structure of the protein (*Putaporntip et al., 2002*). These results follow previously published data demonstrating that antibodies from vivax malaria patients from distinct Brazilian malaria-endemic regions recognized recombinant proteins corresponding to different regions of the *Pv*MSP-1 (*Soares et al., 1997*; *Braga, Fontes & Krettli, 1998*; *Soares et al., 1999*; *Morais et al., 2005*; *Nogueira et al., 2006*; *Ladeia-Andrade et al., 2007*; *Pandey et al., 2010*; *Riccio et al., 2013*; *Versiani et al., 2013*; *Soares et al., 2014*; *Folegatti et al., 2017*).

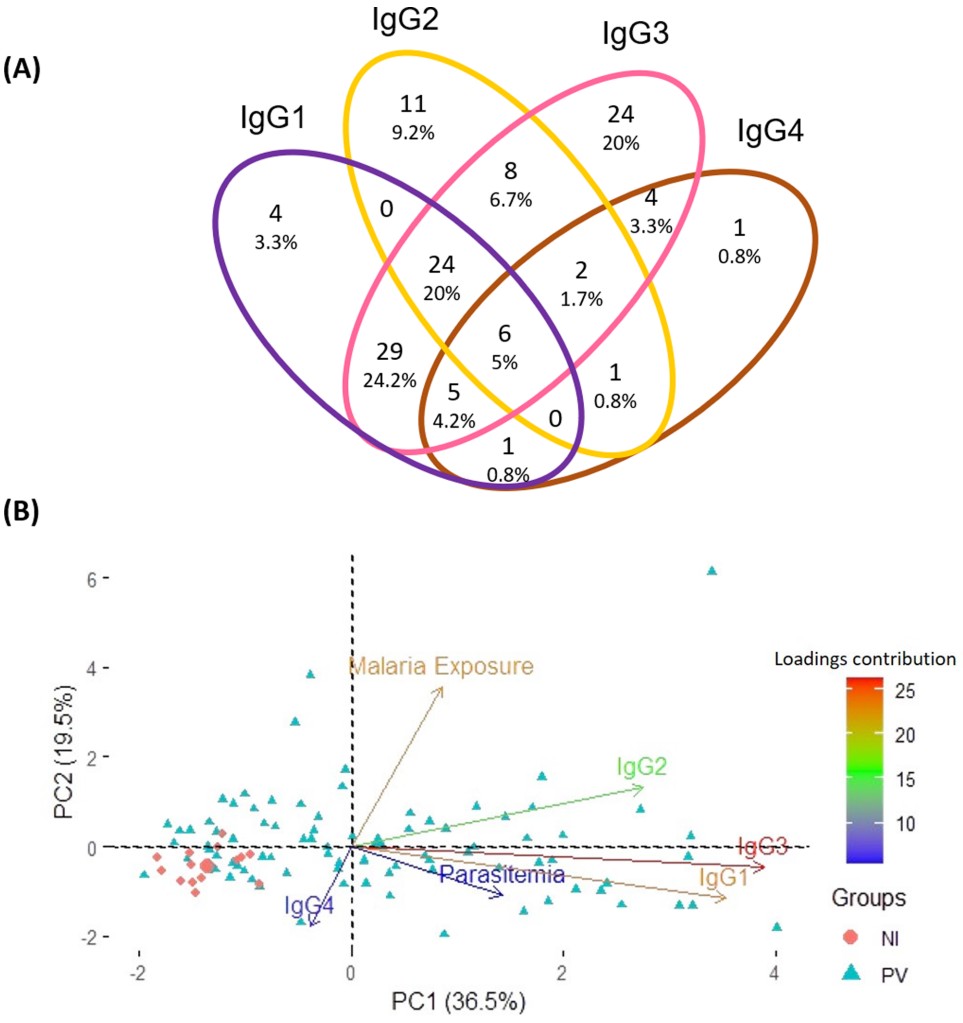

**Figure 5** **Distribution of IgG subclasses and their association with infection and exposure parameters.**
(A) The Venn diagram depicts the number and percentage of *P. vivax*-infected individuals testing positive for each IgG subclass. (B) Principal component analysis illustrates the linkage and variance among IgG subclasses, parasitemia, and malaria exposure in individuals from three distinct groups. NI, non-infected individuals; PV, *P. vivax*-infected individuals.

It is important to note that while the *Pv*MSP-1$_{19}$ subunit is widely recognized and validated as one of the highly immunogenic parts of the protein (*Longley et al., 2020*; *Punnath et al., 2021*), we have discovered through the SPOT-synthesis technique that other most reactive peptides do not align with this region. This outcome is likely due to the conformation-dependent nature of immunogenicity in this particular segment, as many studies use the recombinant expression of this subunit (*Cunha, Rodrigues & Soares, 2001*; *Bueno et al., 2008*; *Rocha et al., 2017*). Our *in silico* analysis showed that p70 and p314 contain promising B epitope sequences. However, serological individual tests with the corresponding soluble synthetic peptides confirmed elevated levels of IgG against p314 in most patients infected with *P. vivax* but, no significant response for p70. These findings

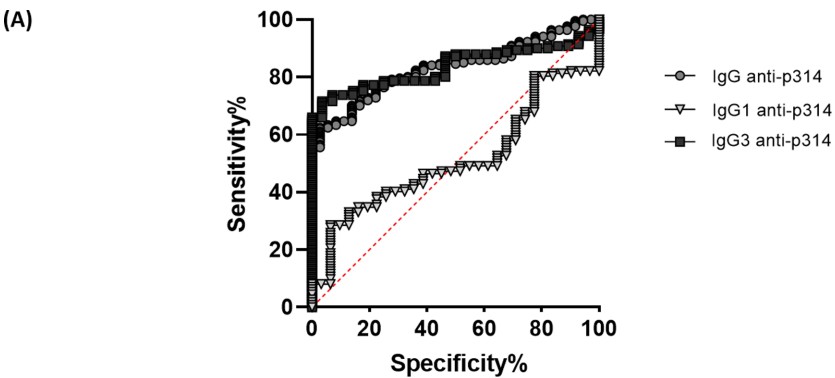

(A)

(B)

|  | Cut-off | Sensibility | Specifity | Area under the ROC curve | P-value |
|---|---|---|---|---|---|
| IgG | > 0.9053 | 76.47 | 77.78 | 0.8322 | <0.0001 |
| IgG1 | > 0.6376 | 49.11 | 48.39 | 0.5084 | 0.8870 |
| IgG3 | > 0.8524 | 77.30 | 78.57 | 0.8354 | <0.0001 |

**Figure 6** **Sensitivity, specificity, and accuracy of IgG, IgG1, and IgG3 responses against p314.** (A) Receiver operating characteristic (ROC) curves depict antibody threshold levels predicting positivity to p314. (B) Comprehensive details of the ROC analysis.

underline the importance of experimental validation following computer-based analysis, as well as the potential of p314 as a diagnostic marker for *P. vivax* exposure. The broad serological reactivity observed for this peptide can be attributed to its amino acid sequence as part of the conserved block 7 within *Pv*MSP-1 (*Putaporntip et al., 2002*). This feature may enhance p314 immune recognition by antibodies derived from patients infected with diverse *P. vivax* strains. Based on our findings, p314 emerged as a putative antigen for distinguishing humoral immune responses between *P. vivax* and *P. falciparum*-exposed individuals in different epidemiological settings. A point that deserves consideration is the fact that p314 is located on the polypeptide subunit 30 of the *Pv*MSP-1, which undergoes proteolytic processing during erythrocyte invasion (*Kauth et al., 2003*; *Babon et al., 2007*). This characteristic raises the prospect of this peptide's presence in the peripheral bloodstream during infection, rendering it a promising target for diagnostic investigation. Further studies are necessary to experimentally validate the potential use of p314 as a biomarker of infection in diverse epidemiological contexts.

While variations in the humoral immune response can occur across diverse endemic settings due to individual and environmental determinants (*Da Silva-Nunes et al., 2008*; *Costa et al., 2020*; *Tayipto et al., 2022*), searching for dependable seroepidemiological markers of malaria has been continuously stimulated. It is a consensus that measuring antimalarial antibodies does not offer a diagnosis of an ongoing *Plasmodium* spp. infection. Instead, on a population scale, specific IgG antibodies may be considered a marker of parasite transmission and exposure, thus serving as an asset for malaria surveillance (*Ubillos, Jiménez & Dobaño, 2018*; *Labadie-Bracho, Van Genderen & Adhin, 2020*; *Longley*

*et al., 2020*; *Longley et al., 2021*; *Kyei-Baafour et al., 2021*; *Villasis et al., 2021*). Considering the acquisition and persistence dynamics of anti-*Plasmodium* antibodies, specific IgG subclass responses may be employed to achieve a more robust prediction of outbreaks and to determine the parasite prevalence in low transmission areas, regardless of their endemic status (*Folegatti et al., 2017*; *Longley et al., 2022*; *Tayipto et al., 2022*). It is well established that pro-inflammatory subclasses IgG1 and IgG3 dominate the antimalarial humoral immune response activating effector cells of the immune system (*Morais et al., 2005*; *Mehrizi et al., 2009*; *Lima-Junior et al., 2011*; *Fantin et al., 2021*; *O'Flaherty, Roe & Fowkes, 2022*). In this study, the plasma levels of IgG3 and IgG1 anti-p314 were significantly elevated in individuals with *P. vivax* infection. Despite the low to moderate parasitemia in patients, possibly attributable to reduced exposure in endemic transmission regions, a positive correlation exists between circulating parasite levels and the IgG1 and IgG3 subclasses, suggesting that parasitemia level influences the serological data obtained. It raises a pertinent question for further investigation: Could levels of anti-p314 cytophilic antibodies in areas of different degrees of transmission, where the population experiences diverse exposure to *P. vivax*, serve as potential markers of endemicity? In this study, IgG3, a shorter half-life immunoglobulin (*Kinyanjui et al., 2007*; *Dechavanne et al., 2017*; *Damelang, SJ & Chung, 2019*; *Chu, Patz Jr & Ackerman, 2021*; *Napodano et al., 2021*; *Ssewanyana et al., 2021*) accounts for most of anti-p314 response and exhibits similar specificity, sensibility, and accuracy to total IgG in distinguishing infected from non-infected individuals. Considering that our work is not characterized as a prospective study, the longevity of the anti-p314 IgG3 response was not determined. Consequently, we cannot definitively state whether specific IgG3 levels would decrease post-treatment, nor p314-specific IgG3 antibody responses may be used to classify individuals recently infected with *P. vivax*.

Accordingly, considering the limited exposure to malaria transmission within the Brazilian Amazon region, our study revealed a diminished IgG2 anti-p314 response, with only 33% of our *P. vivax*-infected subjects exhibiting reactivity (Fig. S2). This finding aligns with the outcomes of a study conducted in Ghana, which established a positive correlation between the prevalence of IgG2 antibodies and individual age (*Dodoo et al., 2008*). Thus, we postulate that the low IgG2 anti-p314 responses may signify a reduced frequency of prior malaria episodes, reflecting the epidemiological context of *P. vivax*-infected patients exposed in the hypo-endemic area of the Brazilian Amazon (*WHO, 2023*).

It is worth mentioning that further studies across diverse epidemiological contexts and geographic regions are necessary before deploying p314-specific antibodies as a serological marker for *P.vivax* exposure. Despite recognizing that sensitivity and specificity values around 75–80% are not yet ideal for serological tests, modification strategies of peptide 314 could circumvent this issue. The biotin incorporation at the ends of the peptide and employing streptavidin for plate affinity could be considered a feasible strategy. This approach would fully expose the epitope region, enhancing IgG recognition (*Perbandt et al., 2007*; *Santamaria, 2020*). Another strategy involves coupling the p314 with a phage-derived icosahedral nanoparticle, which can amplify its exposure, thereby enhancing its recognition (*Brune et al., 2017*; *Bruun et al., 2018*). The importance of incorporating more than one

antigen for the identification of recent exposure (*Longley et al., 2020*; *O'Flaherty, Roe & Fowkes, 2022*) deserves scientific and protocol consideration as well as robust experimental and field validation processes (*Pepe et al., 2001*; *Meibalan & Marti, 2017*; *Foko et al., 2022*). Our study introduces a novel peptide with a facile synthesis, positioning it as a promising candidate for integration into forthcoming serological diagnostic assays, particularly multiplex assays designed for *P. vivax* antigens (*Longley et al., 2022*).

## CONCLUSIONS

In summary, we described for the first time the linear B cell epitope mapping of *Pv*MSP-1 full-length sequence using SPOT-synthesis. We also have demonstrated the potential of antibodies against p314 as biomarkers for *P. vivax* exposure. The characterization of the humoral immune response against p314 represented the initial step in validating the utilization of this peptide as a biomarker for *P. vivax* exposure. Hence, our study proposes a novel and highly immunogenic antigen that can be readily synthesized, serving as a promising candidate to enhance the sensitivity of surveillance tools for detecting *P. vivax* circulation, facilitating the diagnosis and treatment, and aiding in malaria elimination efforts.

## ACKNOWLEDGEMENTS

The authors thank the patients and their families for their contributions to this study. Special thanks to Jamil Silvano de Oliveira for his assistance in the mass spectrometric analysis.

### Funding

This work was supported by Fundação de Amparo à Pesquisa do Estado de Minas Gerais–FAPEMIG (# APQ-00361-16) and Conselho Nacional de Desenvolvimento Científico e Tecnológico–CNPq (# 404365/2016-7, #304334/2019-7). Erika M Braga, Daniella Castanheira Bartholomeu, and Carlos Delfin Chávez Olórtegui are level 1 research fellows from the CNPq/Brazil. The funders had no role in study design, data collection and analysis, decision to publish, or preparation of the manuscript.

### Grant Disclosures

The following grant information was disclosed by the authors:
Fundação de Amparo à Pesquisa do Estado de Minas Gerais–FAPEMIG: # APQ-00361-16.
Conselho Nacional de Desenvolvimento Científico e Tecnológico–CNPq: # 404365/2016-7, #304334/2019-7.

### Competing Interests

Erika M. Braga is an Academic Editor for PeerJ.

## Author Contributions

- Aline Marzano-Miranda conceived and designed the experiments, performed the experiments, analyzed the data, prepared figures and/or tables, authored or reviewed drafts of the article, and approved the final draft.
- Gustavo Pereira Cardoso-Oliveira conceived and designed the experiments, performed the experiments, analyzed the data, prepared figures and/or tables, authored or reviewed drafts of the article, and approved the final draft.
- Ingrid Carla de Oliveira performed the experiments, analyzed the data, prepared figures and/or tables, and approved the final draft.
- Luiza Carvalho Mourão conceived and designed the experiments, analyzed the data, authored or reviewed drafts of the article, and approved the final draft.
- Letícia Reis Cussat performed the experiments, prepared figures and/or tables, and approved the final draft.
- Vanessa Gomes Fraga performed the experiments, authored or reviewed drafts of the article, and approved the final draft.
- Carlos Delfin Chávez Olórtegui analyzed the data, authored or reviewed drafts of the article, and approved the final draft.
- Cor Jesus Fernandes Fontes analyzed the data, authored or reviewed drafts of the article, medical attention of Plasmodium vivax infected patients, and approved the final draft.
- Daniella Castanheira Bartholomeu analyzed the data, authored or reviewed drafts of the article, and approved the final draft.
- Erika M. Braga conceived and designed the experiments, analyzed the data, authored or reviewed drafts of the article, and approved the final draft.

## Human Ethics

The following information was supplied relating to ethical approvals (i.e., approving body and any reference numbers):

Ethics Committee of the National Information System on Research Ethics Involving Human Beings (SISNEP –CAAE: 01496013.8.0000.5149).

## DNA Deposition

The following information was supplied regarding the deposition of DNA sequences:

The PvMSP1 sequence is available at GenBank: PVX 099980.

## Data Availability

The raw measurements are available in the Supplementary Files.

## Supplemental Information

Supplemental information for this article can be found online at http://dx.doi.org/10.7717/peerj.17632#supplemental-information.

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
