# Peer review of "Identification and serological responses to a novel Plasmodium vivax merozoite surface protein 1 (PvMSP-1) derived synthetic peptide: a putative biomarker for malaria exposure"

_PeerJ, doi:10.7717/peerj.17632_

## Round 0.1 · original submission · Major Revisions

I have completed my evaluation of your manuscript. The reviewers recommend reconsideration of your manuscript following major revision. I invite you to resubmit your manuscript after addressing the comments below. When revising your manuscript, please consider all issues mentioned in the reviewers' comments carefully: please outline every change made in response to their comments and provide suitable rebuttals for any comments not addressed. Please note that your revised submission may need to be re-reviewed.

Reviewer 1 ·

Basic reporting

The manuscript entitled “Identification and serological responses to a novel Plasmodium vivax merozoite surface protein 1 (PvMSP-1) derived synthetic peptide: a putative biomarker for malaria exposure” by Marzano-Miranda et al is very well written. The authors showed the potential of antibodies against p314 as biomarkers for diagnostic surveillance P. vivax exposure. In this manuscript the authors identified and characterized the immunodominant peptide of PvMSP1 using a battery of experiments such as peptide array, serological assays, peptide synthesis, and bioinformatic methods) and validated the results appropriately. Overall, the experiments were designed and executed properly, and the results were discussed substantially. In my opinion, the manuscript falls in the journal’s area of interest and is perfectly suitable for publication in its present format.

Experimental design

Methods described with sufficient detail & information to replicate.

Validity of the findings

Conclusions are well stated, linked to original research question & limited to supporting results.

·

Basic reporting

This study titled “Identification and serological responses to a novel Plasmodium vivax merozoite surface protein 1 (PvMSP-1) derived synthetic peptide: a putative biomarker for malaria exposure” investigates the potential of antibodies from the P. vivax exposed patient against specific regions of the Plasmodium vivax Merozoite Surface Protein 1 (PvMSP-1) Sal-I strain as biomarkers for past exposure to P. vivax malaria. The authors have investigated peptides spanning the entire PvMSP-1 sequence to identify the peptide specifically recognized by antibodies from the individuals previously exposed to P. vivax. Authors have identified two major 15-amino acid-long peptides that show very specific interaction with antibodies from patients with known P. vivax exposure. Solid Phase Peptide Synthesis (SPPS) approach was used to synthesize these two peptides (p70 and p314) in a soluble form and subsequently evaluated using an ELISA to determine their effectiveness as biomarkers. Authors have found more specific recognition of p314 by IgG from P. vivax-exposed patients compared to control or P. falciparum. This is well-designed on strong scientific concepts and will be a good addition to the readership of the journal. However, using it as a biomarker at a sensitivity of 76% and specificity of 78% could be debatable. This article could be benefited if the authors could add a few clarifications in the article.
Comments:
1. Authors need to mention that the parasitemia level affects the serological data.
2. It would be interesting to see the sensitivity in the patients from different geographical regions.
3. Why did the authors leave a blank in the parasitemia column in P. vivax at several places? Please add parasitemia in P. falciparum if possible.
4. How this serological method using p314 peptide could benefit although other rapid antigen testing is available with better sensitivity.
5. A few typo errors need to be fixed for example in line 137 PBS as PSB.

Experimental design

I find the experiment design satisfactory.

Validity of the findings

Satisfactory

·

Basic reporting

1. The introduction effectively sets the context by highlighting the malaria transmission in Latin America and the Caribbean, emphasizing the persistence of P. vivax as a major challenge despite overall reduction in transmission. The background information provides relevant statistics and references to establish the significance of the research question. The author should provide the latest malaria data from the WHO 2023 report.
2. The literature review is comprehensive and well-referenced, covering key studies on P. vivax transmission, diagnostic challenges, and the potential of serological assays for malaria surveillance.
3. The structure of the introduction is clear and follows a logical flow from the general context to the specific research focus on PvMSP-1 peptides as serological biomarkers. The introduction effectively transitions from background information to the research objective, providing a clear roadmap for the study.
4. Figures are relevant and well-labeled.
5. No specific details are provided regarding the availability of raw data in the manuscript.

Experimental design

1. Provide more detailed about the enrolled patients, such as age range, gender distribution, and any relevant clinical characteristics.
2. Provide year of sample collection in line 98.
3. Clarify how the 15 patients with P. falciparum infection were selected and their inclusion criteria.
4. Describe the procedures for serum sample collection, storage, and handling to ensure consistency and reproducibility.
5. Discuss the limitations and potential biases associated with predicting B cell linear epitopes using computational tools such as BepiPred 2.0.
6. Consider providing more details on the interpretation and potential limitations of Principal Component Analysis (PCA) for analyzing associations between clinical variables and IgG subclass levels.

Validity of the findings

1. The Results section provides a comprehensive overview of the study's findings, detailing the experimental procedures and outcomes in a clear and structured manner.
2. The authors adeptly interpret the results within the context of existing literature, discussing the significance of antibody recognition patterns and the identification of p314 as a potential biomarker for P. vivax exposure.
3. The integration of in silico peptide validation, ELISA assays, and ROC analysis strengthens the validity of the findings and underscores the practical implications for malaria surveillance and diagnosis.
4. The Discussion section provides insightful analysis and interpretation of the results, elucidating the biological relevance of antibody responses to PvMSP-1 peptides and the potential utility of p314 in serological assays.
5. The authors effectively address the limitations of the study, such as the need for further validation in diverse epidemiological contexts, and propose avenues for future research to enhance the understanding of p314 as a biomarker.

Additional comments

Overall, the study's findings are scientifically sound, and the methodology employed is rigorous, supporting the validity and significance of the results in the context of malaria research and public health interventions.

Reviewer 4 ·

Basic reporting

The manuscript identifies new Plasmodium vivax MSP-1 peptides as potential diagnostic biomarkers of exposure. The authors describe how this work built on previous findings. The article is written in professional, unambiguous English and cites relevant literature to show the context of identifying novel peptides for Pv MSP1.

The manuscript is relevant in the context of malaria elimination programs established in many endemic countries; however, the work could benefit from discussing what makes this work different from other works on Pv MSP1; particularly, the author should point out whether any analysed peptide in this study was located in the region 19 of Pv MSP1. There are plenty of publications on the potential of Pv MSP1 region 19 as a biomarker of exposure, e.g. Longley et al. 2020 (cited by the author).

The manuscript shows the structure required by Peer J and the raw data; however, Figure 4 should be better described to help the interpretation for readers non-familiar with radar charts. I suggest using a more straightforward plot, such as a boxplot. Also, please label Figure 4 properly. The names of the panels described in the captions are not seen in Figure 4.

Experimental design

ROC curve is not described in the method section (lines 204-213). A basic description of this method and interpretation of results should increase the paper's readiness. A brief explanation of what "cut-off" means would make it easy to interpret findings. Is it a Reactivity index cut-off or an ELISA optical density (OD) cut-off?
Please mention what clinical variables the author uses in the PCA analysis (line 208).

Validity of the findings

no comment

---

## Round 0.2 · accepted · Accept

It is a pleasure to accept your manuscript entitled " Identification and serological responses to a novel Plasmodium vivax merozoite surface protein 1 (PvMSP-1) derived synthetic peptide: a putative biomarker for malaria exposure" in its current form for publication in PeerJ.

Reviewer 1 ·

Basic reporting

Pass

Experimental design

With In aim and Scope

Validity of the findings

Novel

Additional comments

NA

·

Basic reporting

This study titled “Identification and serological responses to a novel Plasmodium vivax merozoite surface protein 1 (PvMSP-1) derived synthetic peptide: a putative biomarker for malaria exposure” investigates the potential of antibodies from the P. vivax exposed patient against specific regions of the Plasmodium vivax Merozoite Surface Protein 1 (PvMSP-1) Sal-I strain as biomarkers for past exposure to P. vivax malaria. The authors have investigated peptides spanning the entire PvMSP-1 sequence to identify the peptide specifically recognized by antibodies from the individuals previously exposed to P. vivax. Authors have identified two major 15-amino acid-long peptides that show very specific interaction with antibodies from patients with known P. vivax exposure. Solid Phase Peptide Synthesis (SPPS) approach was used to synthesize these two peptides (p70 and p314) in a soluble form and subsequently evaluated using an ELISA to determine their effectiveness as biomarkers. Authors have found more specific recognition of p314 by IgG from P. vivax-exposed patients compared to control or P. falciparum. This is well-designed on strong scientific concepts and will be a good addition to the readership of the journal.

Experimental design

Satisfactory

Validity of the findings

Satisfactory

Additional comments

The authors have responded to my concerns and I will recommend the article for publication.

·

Basic reporting

The authors have addressed the comments provided earlier. Overall, the study's findings are scientifically sound, and the methodology employed is rigorous, supporting the validity and significance of the results in the context of malaria research and public health interventions. The manuscript can be accepted for publication.

Experimental design

Satisfactory

Validity of the findings

Satisfactory